# Monoclonal antibodies derived from B cells in subjects with cystic fibrosis reduce *Pseudomonas aeruginosa* burden in mice

Malika Hale[1†], Kennidy K Takehara[2†], Christopher D Thouvenel[1], Dina A Moustafa[3], Andrea Repele[1], Mary F Fontana[2], Jason Netland[2], Sharon McNamara[4], Ronald L Gibson[4,5], Joanna B Goldberg[3], David J Rawlings[1,2,5]*, Marion Pepper[2]*

[1]Center for Immunity and Immunotherapies, Seattle Children's Research Institute, Seattle, United States; [2]Department of Immunology, University of Washington School of Medicine, Seattle, United States; [3]Division of Pulmonary, Asthma, Cystic Fibrosis, and Sleep, Department of Pediatrics, Emory University School of Medicine, Atlanta, United States; [4]Cystic Fibrosis Center, University of Washington/Seattle Children's Hospital, Seattle, United States; [5]Department of Pediatrics, University of Washington School of Medicine, Seattle, WA, Seattle, United States

*For correspondence:
drawling@u.washington.edu
(DJR);
mpepper@uw.edu (MP)

†These authors contributed
equally to this work

Competing interest: See page
12

Reviewing Editor: Tomohiro
Kurosaki, Osaka University, Japan

## eLife Assessment

Treatment of *Pseudomonas aeruginosa* (PA) infections is challenging because of intrinsic and acquired antibiotic resistance to most antibiotic drug classes. Therefore, by using donor B cells in subjects with cystic fibrosis who undergo intermittent or chronic airway PA infections, the authors aimed to isolate B-cell receptors against PA virulence factors and examined their biological activities. The data are **solid** and the protective antibodies identified in this study could be **useful** for protection against PA.

**Abstract** *Pseudomonas aeruginosa* (PA) is an opportunistic, frequently multidrug-resistant pathogen that can cause severe infections in hospitalized patients. Antibodies against the PA virulence factor, PcrV, protect from death and disease in a variety of animal models. However, clinical trials of PcrV-binding antibody-based products have thus far failed to demonstrate benefit. Prior candidates were derivations of antibodies identified using protein-immunized animal systems and required extensive engineering to optimize binding and/or reduce immunogenicity. Of note, PA infections are common in people with cystic fibrosis (pwCF), who are generally believed to mount normal adaptive immune responses. Here, we utilized a tetramer reagent to detect and isolate PcrV-specific B cells in pwCF and, via single-cell sorting and paired-chain sequencing, identified the B cell receptor (BCR) variable region sequences that confer PcrV-specificity. We derived multiple high affinity anti-PcrV monoclonal antibodies (mAbs) from PcrV-specific B cells across three donors, including mAbs that exhibit potent anti-PA activity in a murine pneumonia model. This robust strategy for mAb discovery expands what is known about PA-specific B cells in pwCF and yields novel mAbs with potential for future clinical use.

**eLife digest** *Pseudomonas aeruginosa* bacteria currently pose a serious threat to human health worldwide. Not only can these germs cause severe infections, especially in vulnerable individuals, but they are also increasingly resistant to a wide range of antibiotics. New therapeutic approaches are urgently needed.

Based on animal data, some researchers consider monoclonal antibodies to be a promising alternative to antibiotics for treating *Pseudomonas* infections. Produced by immune 'B cells', these tiny molecules are designed to recognize and neutralize germs with incredible specificity and precision. This process starts with an initial exposure to the pathogen, which allows the body to 'learn' how to produce the antibodies against this target. As a result, a dedicated population of B cells carries the genetic sequences necessary to ultimately produce pathogen-specific antibodies.

Initial attempts to treat human *Pseudomonas* infections using monoclonal antibodies have proven safe but largely ineffective. One possible reason is that these trials used mouse-derived antibodies, which may not work as well in people. Typically, successful monoclonal therapies, such as the ones developed for COVID-19, rely on human sequences from individuals who have recovered from infection. In response, Hale et al. set out to design new monoclonal antibodies using a similar approach.

To isolate human *Pseudomonas*-specific immune memory B cells, they turned to blood samples from patients with cystic fibrosis, who frequently experience infections caused by these bacteria. Having collected the genetic sequences that help generate monoclonal antibodies against *Pseudomonas* in these individuals, Hale et al. were able to create human antibodies that could strongly bind to *Pseudomonas aeruginosa* cells and effectively treat infected mice. While further investigation is needed, these findings may help develop new and more effective therapies against these deadly bacteria.

## Introduction

PA is a gram-negative bacteria responsible for significant morbidity and mortality in vulnerable individuals. Treatment is challenging because of intrinsic and acquired antibiotic resistance to most antibiotic drug classes. PA is one of the most common pathogens in severe healthcare-associated infections (*Mancuso et al., 2021*; *Weiner-Lastinger et al., 2020*). Due to the significant mortality caused by multi-drug resistant strains and the lack of alternative therapies, new anti-pseudomonal medicines are urgently needed.

mAbs that bind to key PA virulence factors, such as PcrV, have shown promise in animal models (*Baer et al., 2009*; *Frank et al., 2002*; *Hotinger and May, 2020*; *Sawa et al., 2014*; *Simonis et al., 2023*; *Warrener et al., 2014*). PcrV is a 28 kDa surface protein that forms the distal tip of the type III secretion system required for toxin injection into host cells (*Sato and Frank, 2011*). In prior studies of COVID-19 and malaria, our groups have generated potent anti-pathogen mAbs by sequencing the variable regions of the BCR derived from antigen-specific memory B cells (MBCs) that arise following natural infection in humans (*Hale et al., 2022*; *Rodda et al., 2021*; *Thouvenel et al., 2021*). Therefore, we hypothesized that protective anti-PcrV monoclonal antibodies could be derived from B cells in individuals previously infected with PA.

People with cystic fibrosis (pwCF) experience intermittent or chronic airway PA infections (*Cramer et al., 2023*; *Lund-Palau et al., 2016*). Intermittent infections early in childhood eventually progress to a state of persistent, chronic airway colonization. CF is a multi-organ disease caused by mutations in the gene encoding cystic fibrosis transmembrane conductance regulator (CFTR), an apical epithelial ion channel involved in chloride and bicarbonate transport (*Shteinberg et al., 2021*). Several mechanisms related to the biology of the CF airway have been proposed to explain susceptibility to PA (*Rossi et al., 2021*). Importantly, pwCF do not have intrinsic deficits in adaptive immunity (*Alicandro et al., 2023*; *Ong and Ramsey, 2023*). Class-switched anti-PcrV antibodies are present in CF serum and sputum, at higher concentrations than the general population (*Mauch and Levy, 2014*; *Moss et al., 2001*). However, the PcrV-specific B cells that give rise to antibody-secreting cells had not been previously studied. Here, we show that pwCF have more PcrV-specific circulating B cells compared to non-CF controls. Building on this finding, we utilized single-cell sequencing to identify PcrV-specific paired-chain BCR sequences in pwCF. From

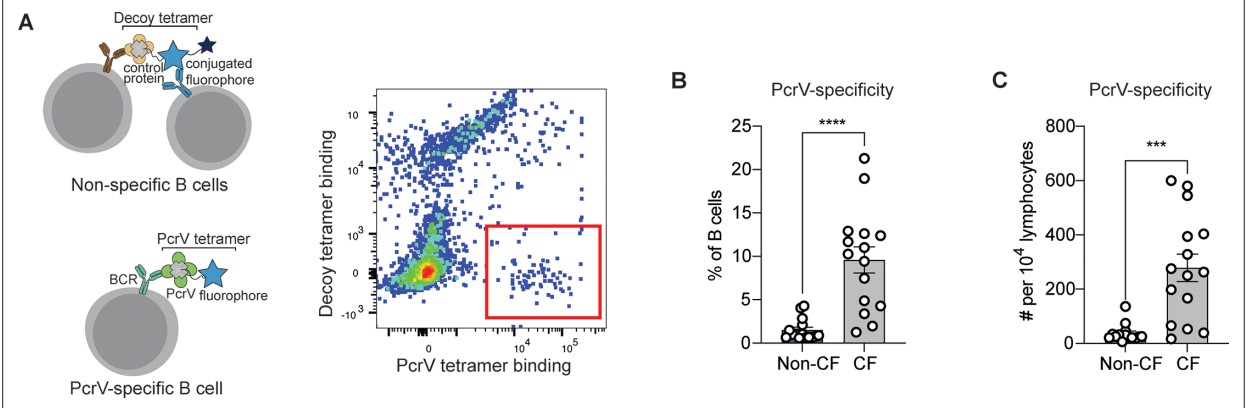

**Figure 1.** PcrV-specific B cells in people with cystic fibrosis (pwCF). (**A**) Schematic of primary human B cells binding to PcrV tetramer reagent and decoy (left). Representative flow plot for B cells after enrichment. Cells binding only to the PcrV tetramer are indicated with the red box. (**B–C**) Percentage (**B**) and number (**C**) of PcrV-specific B cells in pwCF (CF; n=14) vs. control, blood bank donors (non-CF; n=14). Statistics determined by two-tailed Mann-Whitney tests. ***p<0.001, ****pp<0.0001. Error bars represent mean and SD.

The online version of this article includes the following figure supplement(s) for figure 1:

**Figure supplement 1.** Tetramer-specific class-switched B cells in mice immunized with PcrV.

BCR sequences derived from PcrV-specific B cells and MBCs, we generated novel anti-PcrV mAbs, including several that confer robust protection against a virulent strain of PA in an in vivo pneumonia challenge model.

## Results

### PcrV-specific B cells are enriched in CF individuals

We previously described techniques to generate a fluorophore-linked tetramers to isolate antigen-specific B cells (*Krishnamurty et al., 2016*). To identify PcrV-specific B cells, we generated a tetramer using recombinant PcrV protein. In parallel, we also generated a decoy tetramer consisting of irrelevant proteins linked to a tandem fluorophore (*Figure 1A*). To test the tetramer reagent, we immunized C57BL/6 mice with 250 µg purified, recombinant PcrV protein resuspended in Sigma Adjuvant System (SAS, Sigma). Immunization induced an expanded population of class-switched PcrV+ B cells on day 7, demonstrating the specificity of the reagent (*Taylor et al., 2012*; *Figure 1—figure supplement 1*).

We next obtained peripheral blood samples from a cohort of 14 young adults with CF who were followed in CF clinic at Seattle Children's Hospital. As B cells specific for any single antigen are rare, tetramer-bound cells were enriched on an anti-fluorophore magnetic column prior to analysis. After enrichment, PcrV-specific cells made up ≥5% of B cells in 10 out of 14 (71%) of CF donors and demonstrated significant expansion in cell number over non-CF individuals (*Figure 1B*). Similar differences between CF vs. non-CF donors were found when the number of PcrV-specific B cells in each sample was normalized to lymphocyte count (*Figure 1C*).

**Table 1.** Clinical characteristics of subjects from whom PcrV-specific B cell receptor (BCR) sequences were obtained. ETI: elexacaftor/tezacaftor/ivacaftor triple therapy. Chronic infection is defined here as positive *Pseudomonas aeruginosa* (PA) cultures in at least 2 of 3 consecutive years (*Green et al., 2012*; *Rosenfeld et al., 2022*).

| Donor | Age | On ETI? | Lung disease severity | PA infectious history |
|---|---|---|---|---|
| 1 | 33 | Yes | Moderate | Chronic |
| 2 | 19 | Yes | Minimal | Culture-negative at time of blood draw. Culture-positive 6 mo. prior. Previously, 5 y. of negative cultures. |
| 3 | 18 | Yes | Minimal | Chronic |

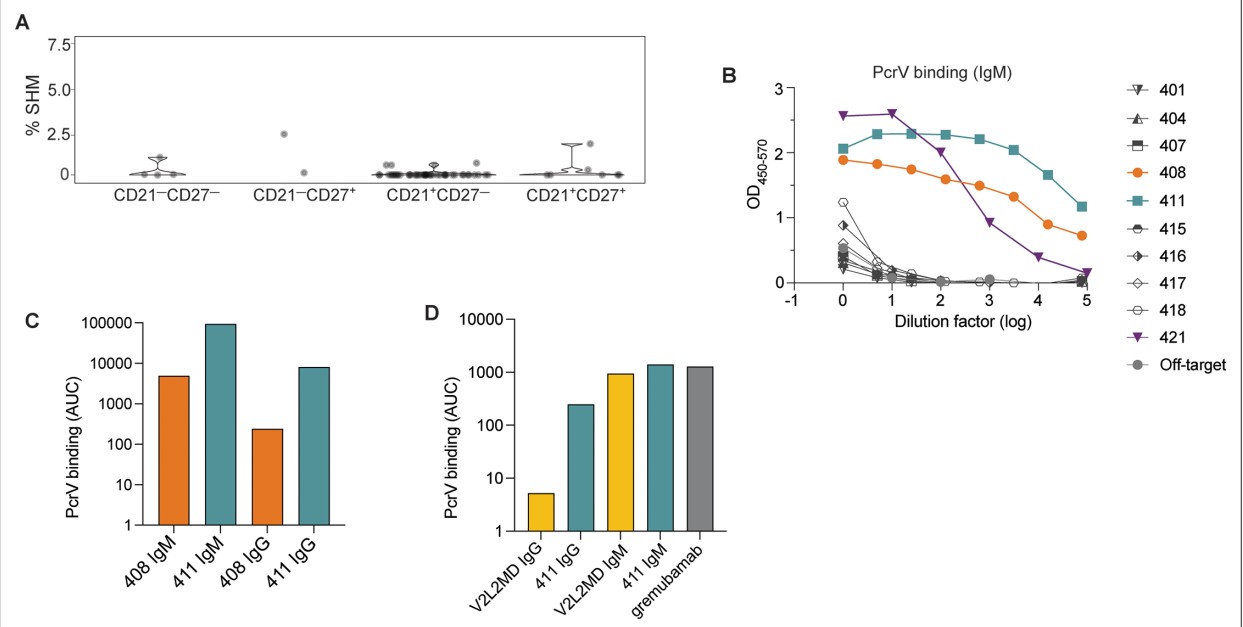

**Figure 2.** Generation of anti-PcrV mAbs derived from B cells isolated from a *Pseudomonas aeruginosa* (PA)-infected cystic fibrosis (CF) donor. (**A**) Percentage of somatic hypermutation (SHM) detected in B cell receptor (BCR) sequences from cells of the indicated phenotype for CF donor 1, shown as violin plot. (**B**) ELISAs showing PcrV binding using supernatants derived from 293T cells transfected with IgM expression plasmids containing the indicated BCR sequences. (**C**) Area under the curve (AUC) for representative ELISAs of purified antibodies 408 and 411 expressed alternatively as IgM vs. IgG in a single experiment. (**D**) AUC for representative ELISA of purified mAbs V2L2 and 411 expressed, alternatively as, IgG vs. IgM; and for the commercially sourced, clinical, bi-specific mAb, gremubamab.

The online version of this article includes the following figure supplement(s) for figure 2:

**Figure supplement 1.** Summary of data generated by single-celB cell receptor l BCR sequencing.

### Generation of mAbs from IgM⁺ B cells derived from a CF donor with PA

We next sought to isolate and sequence B cell receptors from PcrV-specific B cells in order to generate monoclonal antibodies. PBMCs were isolated from a blood sample from a CF donor (Donor 1), who was chronically infected with PA based on at least 2 PA + oropharyngeal cultures in three consecutive years collected as part of routine care (*Green et al., 2012*; *Mogayzel et al., 2014*; *Rosenfeld et al., 2022*) (see *Table 1* for clinical characteristics). Following single-cell sorting of PcrV-specific cells (*Figure 2— figure supplement 1A*), we performed paired chain sequencing of the heavy and light chain variable regions. Our FACS sorting method allows single-cell surface phenotyping data to be linked to the BCR sequence for individual cells (*Krishnamurty et al., 2016*). Interestingly, only two PcrV-specific B cells sequenced in Donor 1 expressed the canonical MBC surface markers (CD21⁺CD27⁺) (gating strategy shown in *Figure 2—figure supplement 1B*). Consistent with these observations, BCR sequencing of PcrV-specific B cells revealed predominantly germline sequences with little or no evidence for somatic hypermutation (*Figure 2A*).

We next generated mAbs that employed the BCR variable regions from 10 tetramer-bound B cells isolated from Donor 1. As the surface-expressed isotype of all these PcrV-specific B cells was IgM (*Figure 2—figure supplement 1C*), we first expressed the BCR sequences as pentameric IgM antibodies. When tested for binding to recombinant PcrV in a plate-bound ELISA assay, supernatants collected from cells co-transfected with heavy, light, and J chain plasmids (to make pentameric IgM) exhibited strong binding to PcrV for 3 of 10 CF-derived BCRs (denoted BCRs 408, 411, and 421) (*Figure 2B*). The light chain for BCR 421 contains several mutations in CDR1. Despite having germline sequences, mAbs from clones 408 and 411 had high affinity as measured by ELISA. As clinical application would likely require large-scale production in a similar system, we chose to move forward with the best binders that also were most efficiently expressed in the 293T cell line: 408 and 411 (data not shown).

We have previously found that for mAbs that target divergent pathogens, including SARS-CoV-2 and *Plasmodium falciparum*, multimerized antibodies (eg. pentameric IgM) have enhanced binding and protection properties in vitro (*Hale et al., 2022*; *Thouvenel et al., 2021*). However, nearly all current monoclonal antibody therapeutics employ the IgG1 isotype (*McConnell, 2019*). Therefore, we next sought to compare the activity of these mAbs in both the IgM and IgG1 formats, and generated expression plasmids containing the heavy chain variable regions upstream of the gamma1 constant region to enable expression as IgG1 mAbs. After co-transfection with the paired light chain (and co-expressed J chain for IgM production) and antibody purification, we were able to compare binding to recombinant PcrV for BCR sequences 408 and 411 when expressed as IgG *vs.* IgM mAbs (*Figure 2C*). CF BCRs 408 and 411 exhibited binding to PcrV as both IgM and IgG mAbs; with higher binding as IgM, reflecting the avidity of IgM multimers (*Hale et al., 2022*; *Thouvenel et al., 2021*).

One challenge in studying candidate human PcrV mAbs is the lack of readily available comparators. V2L2MD is a heavily engineered anti-PcrV mAb that was generated by phage display binding optimization from a precursor candidate identified by hybridoma screening from PcrV-immunized, human-variable region transgenic mice (*Warrener et al., 2014*). The clinical candidate bi-specific, gremubamab, consists of the paired heavy and light chain variable regions of V2L2MD fused to an anti-polysaccharide (Psl) single chain variable fragment (scFv) and the human IgG1 constant region (*Chastre et al., 2022*; *DiGiandomenico et al., 2014*). To prepare for in vivo protection assays, we introduced gremubamab's anti-PcrV variable regions (V2L2MD) into expression plasmids using the same strategies we have employed for expressing CF BCRs. This approach enabled us to produce and purify positive control human V2L2MD IgG and IgM mAbs in parallel with our candidate CF BCR-derived mAbs. To confirm specificity, we measured binding to recombinant PcrV for purified IgG and IgM mAbs produced in parallel from the newly cloned V2L2MD and the best candidate (411), together with commercially sourced gremubamab (*Figure 2D*). As IgM mAbs, 411 and V2L2MD exhibited similar binding to PcrV, on par with the commercially-sourced gremubamab.

## Anti-PcrV mAbs derived from CF B cells protect mice from PA pneumonia

Next, we used a challenge model of pneumonia to test the anti-PA activity of candidate mAbs (*Figure 3A*). Mice treated with a single, 20 µg intranasal dose of anti-PcrV IgG (408 and 411) exhibited an ~2 log reduction in burden of bacteria in the lungs at 24 hr in comparison with an off-target control mAb or vehicle (PBS) only (*Figure 3B*). The CF-derived mAbs performed equivalently to the positive control V2L2MD mAb.

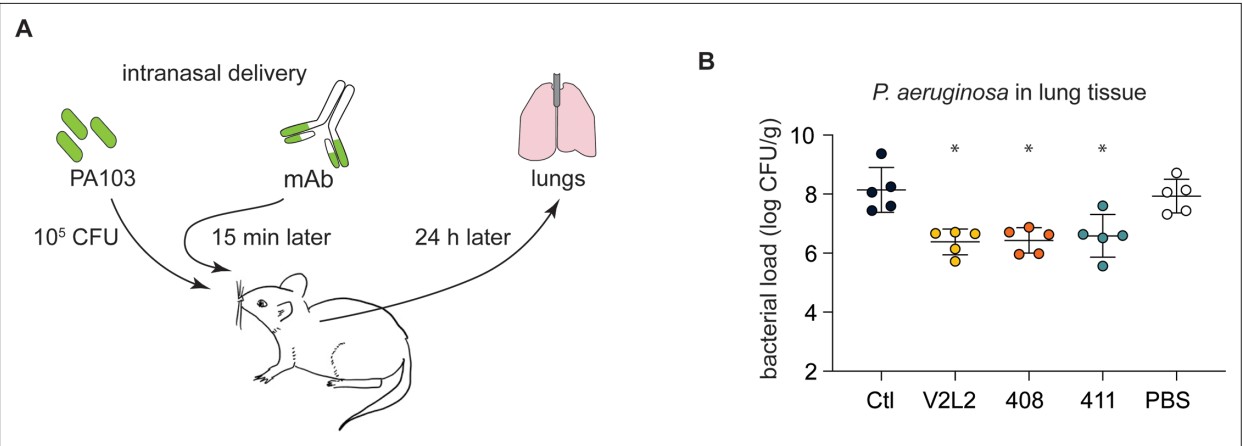

**Figure 3.** People with cystic fibrosis (pwCF)-derived, germline, anti-PcrV-specific mAbs exhibit robust anti-*Pseudomonas aeruginosa* (PA) activity in an in vivo mouse pneumonia model. (**A**) Schematic illustrating the experimental PA infection and mAb delivery protocol. (**B**) Bacterial load in mouse lungs at 24 h post-infection for mice that received 20 µg intranasal dose of off-target, control IgG1 mAb (Ctl), indicated anti-PcrV mAbs, or diluent alone (PBS). Error bars represent mean and SD. Asterisks show significance in Dunn's test versus animals treated with diluent only (PBS); *p<0.05.

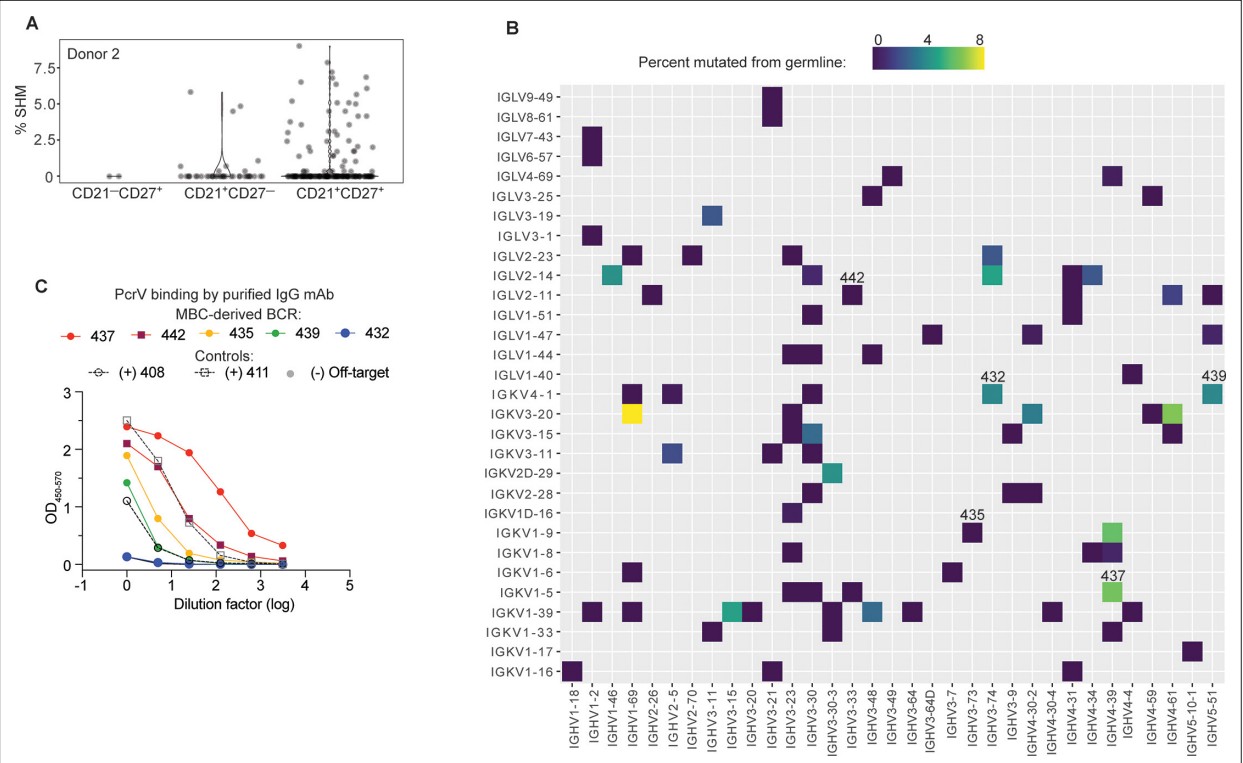

**Figure 4.** High affinity anti-PcrV mAbs derived from memory B cells isolated from cystic fibrosis (CF) Donor 2. (**A**) Somatic hypermutation (%SHM) rates in B cell receptor (BCR) sequences from individual B cells with the indicated surface phenotype isolated from CF donor 2, shown as violin plot. Each circle represents a heavy or light chain sequence from a singly-sorted cell. (**B**) Heatmap showing paired heavy (x-axis) and light (y-axis) V genes for 79 MBCs in donor 3. The color gradient depicts SHM for the heavy chains in each pairing. Clone numbers for BCRs subsequently expressed as mAbs are included above their corresponding box. (**C**) PcrV binding for purified mAbs generated from five MBC BCRs (colored lines) screened in a single ELISA. The two Donor 1-derived mAbs with in vivo protective activity (408 and 411) are used as benchmarks for relative binding activity (dashed lines). The off-target control (anti-SARS-CoV-2 RBD) line appears hidden because it overlaps with the blue line (432).

The online version of this article includes the following figure supplement(s) for figure 4:

**Figure supplement 1.** B cell receptor (BCR) sequencing of PcrV-tetramer-specific B cells derived from three cystic fibrosis (CF) donors.

**Figure supplement 2.** Transfectant supernatant screen of 12 MBC-derived monoclonal antibodies (mAbs).

## Anti-PcrV mAbs derived from MBCs from pwCF exhibit in vivo protection

The above findings demonstrate that isolation of PcrV-specific B cells from a chronically infected CF donor enabled the rapid discovery of two protective mAbs. In our earlier studies of other pathogens, antigen-specific MBCs had been excellent sources for high-affinity, protective BCR sequences (*Rodda et al., 2021*; *Thouvenel et al., 2021*). Therefore, we hypothesized that higher affinity, anti-PcrV mAb might be isolated from antigen-experienced, somatically hypermutated, PcrV-specific MBCs. We obtained peripheral blood samples from two additional pwCF: Donor 2, who had tested culture positive for PA for the first time 6 mo prior to sample collection and was negative on most recent testing, and Donor 3, who like Donor 1, was considered to be chronically infected at the time of blood sample (*Table 1*; *Green et al., 2012*; *Rosenfeld et al., 2022*). Using improved methods that enhanced the efficiency and depth of sequencing (manuscript in preparation as Thouvenel et al.), we performed single-cell BCR sequencing of PcrV-specific B cells from each donor. Sequences derived from PcrV-specific cells identified by surface phenotype as likely MBCs (CD21[+]CD27[+]) carried an average somatic hypermutation load of 2.5% in Donor 2 (*Figure 4A*) and <1% in Donor 3 (*Figure 4—figure supplement 1A*).

There was significant breadth in V gene usage within and between donors (*Figure 4—figure supplement 1B*). The small number of cells analyzed, compared to B cell repertoire studies that employ bulk-sorted populations, limits the interpretation of our data. However, a strength of singly sorted B cell sequencing is the ability to obtain paired chain information. The heavy chain IGHV3-23

is used, with mutations, in the V2L2MD antibody, and is a very common VH gene segment in most human B cell repertoire studies (*Rubelt et al., 2016*). Accordingly, IGHV3-23 was also well represented in this dataset. Notably, none of the PcrV-specific B cells sequenced for our study paired VH3-23 heavy chains with IGKV1-6, the V2L2MD light chain variable gene (*Figure 4—figure supplement 1C*), and IGKV1-6 was infrequent in our dataset. Heavy- and light-chain pairings for CD21$^+$CD27$^+$ cells are shown in *Figure 4B*.

To pursue our goal of identifying potential protective mAbs among MBCs, 12 of these BCRs were randomly selected for cloning into IgG1 expression plasmids. Supernatants from transfected cells were screened for binding to PcrV (*Figure 4—figure supplement 2*). BCRs 435 and 442 were the most striking binders in the transfection screen. Because our initial screen did not adequately control for antibody concentration in the supernatant, antibodies that are less efficiently expressed by 293T cells might appear to be poor binders. Based on performance in the screen or for reasons of special interest (for example, the raw heavy chain sequence that became BCR clone 439 included the IGHA constant region that defines the IgA antibody isotype, the predominant isotype in respiratory secretions), 5 MBC-derived BCRs were chosen for production as purified IgG mAbs. When this panel of purified mAbs was evaluated at matched concentrations (*Figure 4C*), 2 of 5 of MBC-derived mAbs matched or exceeded PcrV binding exhibited by the protective, donor 1 mAb, 411. Furthermore, 4 of 5 matched or exceeded the other protective donor-1 derived protective mAb, 408 IgG.

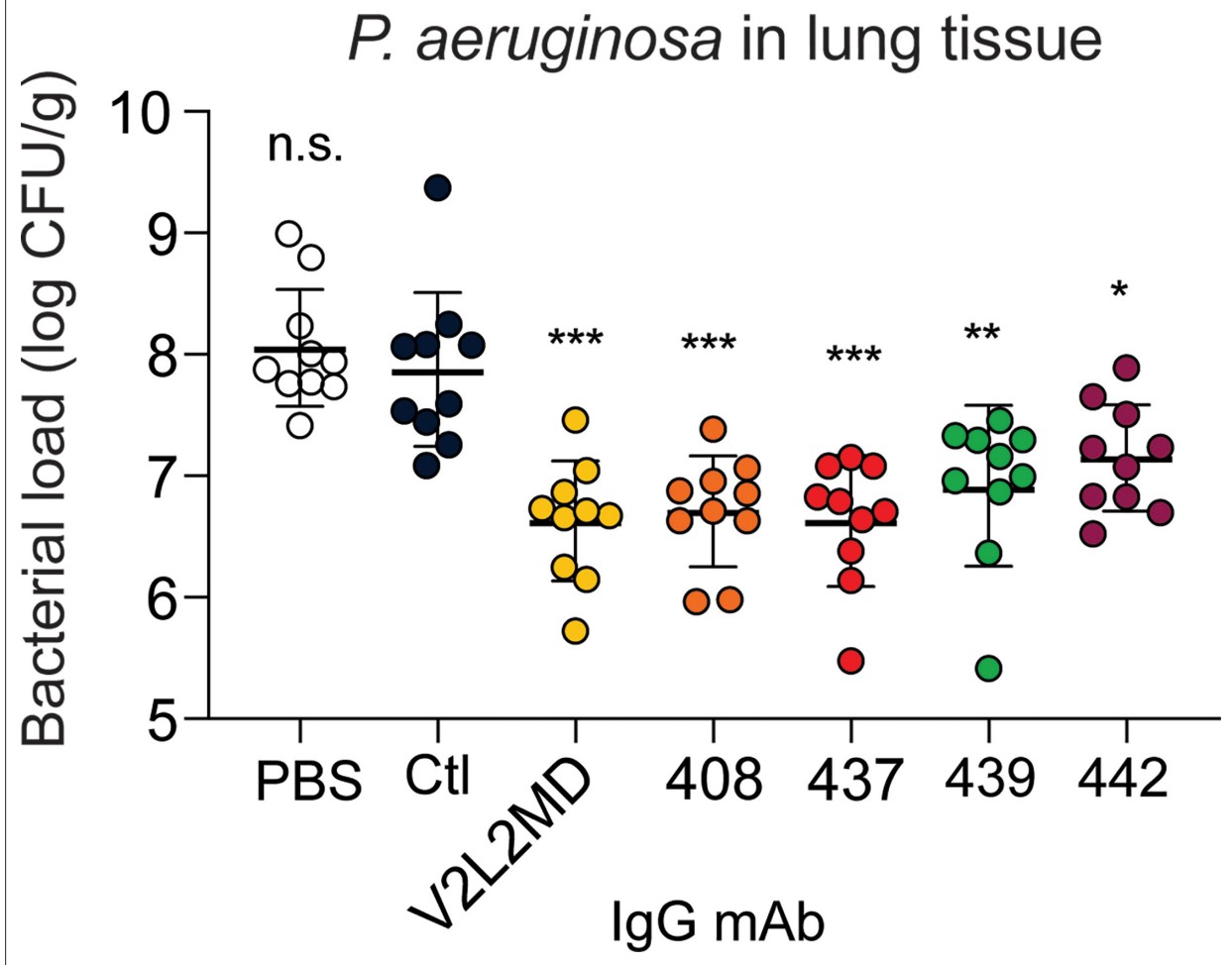

**Figure 5.** mAbs derived from cystic fibrosis (CF) memory B cell (MBC) B cell receptor (BCR) sequences control *Pseudomonas aeruginosa* (PA) infection in a murine pneumonia model. Lung bacterial load for mice treated intranasally with 20 µg of the indicated anti-PcrV mAb, or vehicle only (PBS). Combined data from two independent experiments is shown (n=10 mice per condition). Error bars show mean and SD. Asterisks show significance in Dunn's test versus animals treated with the off-target control antibody (Ctl); *p<0.05, **p<0.01, ***p<0.001, n.s.: not significant.

Next, we tested a moderate-affinity MBC-derived mAb (439) in addition to the two best-performing MBC-derived mAbs (437 and 442) for analysis of in vivo protection. Strikingly, all three CF MBC-derived mAbs including 437 (originally derived from an IgG MBC), 439 (derived from an IgA MBC), and 442 (derived from an IgM MBC) achieved significant reductions in bacterial burden at 24 hr when compared to an off-target control mAb or saline alone (*Figure 5*).

In summary, we obtained 175 unique paired heavy-light chain BCR sequences of PcrV-binding B cells from 3 donors with CF. After expressing 20 BCRs as human mAbs, we confirmed at least moderate in vitro PcrV binding for the majority screened (12 of 20, 60%), and high-affinity binding for 7 (35%). Of the 5 mAbs tested in an established in vivo model of PA pneumonia, all produced significant reductions in lung bacterial burden. In comparison to standard methods for antibody discovery using immunized mice and screening hundreds to thousands of hybridomas, our approach was more efficient. In addition, our results reveal a diverse PcrV-specific repertoire for PA-specific B cells in pwCF.

## Discussion

In the face of rising antibiotic resistance by pathogenic bacteria, new antimicrobial treatment modalities are needed. Unfortunately, relative to the projected need, few novel antimicrobials are under active pharmaceutical development (*Theuretzbacher et al., 2020*). Antibacterial mAbs could fill the widening gap as alternatives, or adjuncts, to traditional antibiotics. Because mechanisms of intrinsic resistance already limit therapeutic options, multidrug-resistant PA is especially difficult to treat (*Yaeger et al., 2021*). Efforts to develop mAbs targeting PcrV, a critical component of the PA toxin injection apparatus, are bolstered by strong evidence for antibody-mediated protection in animal models (*Goure et al., 2005*; *Moriyama et al., 2009*; *Sawa et al., 1999*; *Sécher et al., 2019*). Unfortunately, two engineered antibody-like drug candidates (a Fab fragment and a bi-specific mAb) derived from mice failed to achieve efficacy in human trials (*Chastre et al., 2022*; *François et al., 2012*; *Jain et al., 2018*; *Yaeger et al., 2021*). New approaches are needed to identify fully human mAbs with robust anti-PA protective activity, reduced immunogenicity, and improved pharmacokinetics.

Traditional methods of mAb development begin with immunization of mice with a recombinant protein. To address the immunogenicity of the murine constant region, mouse-derived antibodies must then be iteratively re-humanized in vitro and/or modified into antibody fragments (Fab) which lack effector functions and are subject to rapid renal clearance. Although fully in vitro strategies like phage display can enable the use of human variable domains (*Tavernier et al., 2016*; *DiGiandomenico et al., 2012*), the high-throughput screening processes are vulnerable to bottleneck effects and drift, and lack key features of in vivo B cell development, such as selection against autoreactivity.

Here, we generated multiple anti-PcrV mAbs directly from the BCR variable regions of antigen-specific B cells derived from CF donors with intermittent (past) or persistent Pa infection. For practical reasons, we tested only a subset of paired antibody sequences obtained. However, our antibody discovery strategy was extremely efficient, requiring expression of very few (<12) BCRs to yield protective anti-pseudomonal mAbs. Strikingly, in in vivo challenge studies, 4 of 5 mAbs directly cloned from human B cells achieved control of PA. While assessment of lung bacterial burden requires sacrifice of the treated mice, doses achieving similar reductions in lung bacterial burdens have resulted in 100% survival in cohorts of mice in prior studies of V2L2MD (*Warrener et al., 2014*). Our findings employing BCR sequences from multiple isotypes are consistent with contemporaneous work by *Simonis et al., 2023* showing protection in neutropenic mice by three anti-PcrV mAbs discovered by single-cell BCR sequencing of IgG-expressing B cells from two CF donors.

Notably, the anti-PcrV mAbs antibodies isolated using our approach matched the in vivo performance of V2L2MD IgG, the highly engineered PcrV-specific binding component of the clinical bi-specific antibody, gremubamab. The anti-PcrV antibody (V2L2MD) was effective in animal models, but the bi-specific antibody containing this anti-PcrV binding moiety failed to achieve the targeted efficacy outcomes in Phase 2 (*Chastre et al., 2022*; *DiGiandomenico et al., 2014*). As noted above, V2L2MD IgG was originally derived from antibody sequences elicited by immunizing human-variable region transgenic mice with purified recombinant PcrV and subsequently optimized via phage display (*Warrener et al., 2014*). Importantly, mAbs derived directly from human B cells have potential theoretical advantages in human use. Compared to animal-derived anti-pseudomonal candidates like

gremubamab, human MBC-derived mAbs are expected to exhibit reduced immunogenicity *Strand et al., 2017* and manifest fully human effector functions. Whether such antibodies will be more effective in the clinical setting, however, remains to be determined.

While the goal of this study was to generate novel protective mAbs for potential therapeutic use, our findings also offer new insight into the cellular sources of the humoral response to PA in pwCF, expanding on studies which have thus far been limited to the measurement of secreted antibodies (*Mauch and Levy, 2014*). As we hypothesized, given the uniquely high frequency of PA infections in the CF population, circulating PcrV-specific B cells in CF individuals were expanded compared to non-CF individuals. While the ability to generate protective mAbs ex vivo from CF B cells demonstrates that B cells with the technical capacity to produce functional anti-PA antibodies are present in the peripheral blood, and recent work has shown that anti-PcrV antibodies capable of neutralizing PA in vitro are present in some pwCF (*Simonis et al., 2023*), multiple in vivo challenges may limit PA clearance. Significant future work will be required to fully characterize the nature and dynamics of PA-specific B cells in CF individuals. Better understanding of the endogenous anti-PA response might also reveal insight that could enhance future therapeutic antibody products, enabling optimization of isotype, glycosylation, delivery site, or other features.

We have previously shown that protective mAbs may be built from antigen-specific MBCs that target the parasite *Plasmodium falciparum* and the virus SARS-CoV-2 (*Hale et al., 2022*; *Rodda et al., 2021*; *Thouvenel et al., 2021*). Besides eliminating the humanization requirement and concern for residual immunogenicity, human MBC-derived mAbs are the product of evolved B cell development and maturation processes that encompass tremendous diversity and include selection against autoreactivity and, for cells that have exited GCs, pre-optimized binding to antigen via affinity maturation. When combined with the results of our present study, human antigen-specific B cells may be broadly considered as an underutilized high-yield resource in the critical endeavor to discover new treatments for infectious diseases.

## Materials and methods

**Key resources table**

| Reagent type (species) or resource | Designation | Source or reference | Identifiers | Additional information |
|---|---|---|---|---|
| Strain, strain background (*Pseudomonas aeruginosa*) | PA103 | ATCC | Cat#:29260 | sputum isolate |
| Cell line (human) | HEK-293T (epithelial-like, kidney) | ATCC | Cat#:CRL-3216; RRID:CVCL_0063 | Cell line transfected for antibody production |
| Transfected construct (human) | Antibody expression plasmids (plasmid) | Addgene | RRID:Addgene_183702 (kappa light chain);RRID:Addgene_183703 (lambda light chain); RRID:Addgene_80795 (IgG1 heavy chain) | Deposited in Addgene by Dr. Hedda Wardemann |
| Biological sample (human) | PBMC | Seattle Children's Cystic Fibrosis Clinic | | Collected from individual donors under IRB approval SCH#10325 |
| Antibody | HRP-conjugated anti-human IgG; IgM (goat polyclonal) | SouthernBiotech | Cat#:2040–05 (IgG); Cat#: 2020–05 (IgM) | ELISA IgG (1:3000); IgM (1:1500) |
| Antibody | HRP-conjugated anti-human IgA (goat polyclonal) | Thermo Fisher Scientific | Cat#:A18781 | ELISA (1: 1:1500) |
| Recombinant DNA reagent (*Pseudomonas aeruginosa*) | pET16b PcrV (plasmid) | *Sawa et al., 1999* | | Plasmid encoding histidine-tagged PcrV; gift from Dr. Timothy Yahr |
| Peptide, recombinant protein | Streptavidin conjugated to fluorophore | Agilent | Cat#:PJ27S (SA-APC); Cat#:PJRS25 (SA-PE) | For labeling of protein tetramer |
| Commercial assay or kit | In-Fusion HD Cloning | Takara Bio | Cat#:639650 | |

*Continued on next page*

*Continued*

| Reagent type (species) or resource | Designation | Source or reference | Identifiers | Additional information |
|---|---|---|---|---|
| Commercial assay or kit | EZ-Link Sulfo-NHS-LC Biotinylation Kit | Thermo Fisher | Cat#:21435 | |
| Commercial assay or kit | SMART-seq v4 | Takara Bio | Cat#:634770 | cDNA synthesis |
| Chemical compound, drug | gremubamab | Invitrogen/Thermo Fisher Scientific | Cat#:MA5-42275 | |
| Software, algorithm | IgBlast | *Ye et al., 2013* | | |
| Software, algorithm | IGMT/HighV-Quest | *Alamyar et al., 2012* | | |
| Software, algorithm | Prism | GraphPad | | Version 9.5 |
| Software, algorithm | TRUST4 | *Song et al., 2021* | | |
| Other | Amicon Ultracentrifugal Filter | Sigma Aldrich | Cat#:UFC9050 | For concentration and buffer exchange of purified mAbs |
| Other | HiTrap Protein G HP purification column | Cytiva/Thermo Fisher Scientific | Cat#:45-000-053 | For purification of IgG from concentrated transfectant supernatant |
| Other | POROS CaptureSelect IgM-XL Affinity Matrix | Thermo Fisher Scientific | Cat#:2812892005 | For purification of IgM from concentrated transfectant supernatant |

## PBMC and serum collection

Under protocols approved by the Seattle Children's Institutional Review Board (SCH#10325), CF donors were recruited from patients receiving care that day at Seattle Children's Cystic Fibrosis Clinic in March-June 2019 and April-July 2021. Individuals who declined to participate or were unable to safely donate at least 30 mL of blood were excluded. Blood was transported at room temperature to the University of Washington Department of Immunology and processed within 4 hr of collection. Serum was collected by centrifugation at 1500 g for 10 min, and then frozen at –80° C. Peripheral mononuclear blood cells (PBMCs) were collected using SepMate-50 PBMC Isolation Tubes (STEM-CELL Technologies) and frozen slowly at –80° C before transfer to liquid nitrogen for long-term storage. Non-CF donor samples were provided by BloodWorks Northwest from regular blood donors or from non-mobilized healthy donors through the Fred Hutch Hematopoietic Cell Procurement and Processing core.

## Tetramer production

Recombinant his-tagged PcrV was first expressed and isolated from DH5α pET16b pcrV (kind gift from Dr. Timothy Yahr, University of Iowa; *Sawa et al., 1999*) and purified from filtered bacterial supernatant on a His affinity column. To enable tetramerization, PcrV was biotinylated using EZ-Link Sulfo-NHS-LC Biotinylation Kit (ThermoFisher). The production of antigen-specific B cell tetramer reagents has been previously described (*Krishnamurty et al., 2016*). Briefly, biotinylated PcrV was co-incubated with streptavidin-fluorophore (SA-PE or SA-APC; Agilent). Decoy tetramers are produced by tetramerization of an irrelevant protein with a matched conjugated fluorophore (e.g. PE-Cy7 for use in experiments requiring the PE-conjugated PcrV tetramer).

## Identification of PcrV-specific cells

Our methods for isolating antigen-specific B cells using tetramer reagents are described in detail elsewhere (*Taylor et al., 2012*). Flow cytometry was performed on a BD LSR II (Becton Dickinson). For BCR sequencing experiments, single cells were sorted into a 96-well-microplate using a BDFACS Aria II that was contained within a biosafety cabinet.

## BCR sequencing

cDNA was amplified from singly sorted B cells using SMART-Seq v4 (Takara Bio) at half reaction volumes. Initial BCR sequencing for donor 1 followed protocols we have previously described in detail

(*Hale et al., 2022*; *Rodda et al., 2021*; *Thouvenel et al., 2021*). Briefly, a single, multiplex PCR was performed for each B cell using a universal primer for the template switch region and pooled constant region primers for the μ, γ, α, κ, and $\lambda$ constant regions. Amplicons were then purified and sequenced by Sanger sequencing. For donors 2 and 3, single-cell BCR sequencing was performed using a protocol adapted for MiSeq (Illumina). Alignment of all trimmed sequences was performed using both TRUST4 (*Song et al., 2021*) and IGMT/HighV-QUEST (*Alamyar et al., 2012*; *Ehrenmann and Lefranc, 2011*). Rare conflicts (e.g. differences in reported %SHM) were resolved by review of the raw sequence data and individual analysis in IgBlast (*Ye et al., 2013*).

## BCR cloning

Our methods for cloning BCR variable region sequences into antibody expression plasmids were described previously (*Hale et al., 2022*; *Rodda et al., 2021*; *Thouvenel et al., 2021*). Briefly, each light chain was cloned into vectors of its isotype, κ or $\lambda$, following the manufacturer's protocol for in-fusion cloning (Takara Bio). All heavy chains were similarly cloned into IgG1 and IgM plasmids in parallel. Concordance with the parental cDNA was confirmed by Sanger sequencing of the cloned plasmids.

For the V2L2MD IgG and IgM mAbs, heavy and light chain sequences were synthesized as a gBlock (IDT) by introducing mutations to match the amino acid sequence for the anti-PcrV VH/VL in gremubamab (INN 10909) in the closest human VH and VL nucleotide sequences. The resultant V(D)J sequences were then synthesized as a gBlock (IDT) and then cloned into expression plasmids using the same methods described for generation of our human B cell BCR sequences.

Whole-plasmid sequences were obtained from a subset of plasmids as a quality control measure (Primordium Labs).

## Production of mAbs

IgG mAbs were produced in HEK-293T cells (ATCC) by co-transfection of heavy- and light-chains in polyethylenimine as previously described (*Thouvenel et al., 2021*). Production of IgM mAbs was carried out as described in Hale et al., using the same human J chain plasmid. For initial screens, supernatant was harvested at 4 d post-transfection and concentrated and buffer-exchanged into PBS using 50,000 MWCO Millipore Amicon Ultra-15 Centrifugal Filter units (Thermo Fisher Scientific). Purification was carried out following the manufacturer's instructions on HiTrap Protein G HP purification column for IgG mAbs (GE Healthcare), and a POROS CaptureSelect IgM-XL Affinity Matrix Column (Thermo Fisher Scientific) for IgM. Antibodies were concentrated and buffer-exchanged into phosphate buffered saline to at least 1 mg/mL, and then stored at –80° C in 120–200 μL aliquots.

## ELISAs

To prepare for antigen-specific antibody ELISAs, recombinant PcrV was prepared as described above and diluted to 2 μg/ml in PBS and incubated on high-binding 96well plates (Corning) overnight at 4 °C. Plates were then washed thoroughly with PBS and 0.05% Tween 20 (PBS-T). Next, non-specific interactions were blocked using 200 μl/well of PBS-T with 3% bovine serum albumin (3%) at room temperature for 1–3 hr. Samples of interest were serially diluted in PBS-T immediately prior to use. Dilution series for each sample were incubated on washed plates for 2 hr at room temperature. After thorough washing with PBS-T, bound antibodies were detected by incubation for 1 hr at room temperature with HRP-conjugated goat anti-human IgG (diluted 1:3000 in PBS-T; SouthernBiotech), IgM (1:1500; SouthernBiotech), or IgA (1:1500; ThermoFisher), washed again, and developed using 1×3,39,5,50-tetramethylbenzidine (Invitrogen) and 1 M $H_2SO_4$. OD was measured on a SpectroMax i3X (Molecular Devices, San Jose, CA) at 450 and 570 nm, and $OD_{450-750}$ was analyzed in Prism (v9.5; GraphPad). When un-purified transfectant supernatants were used, ELISAs for total IgM or IgG were also performed in parallel using human uncoated IgM or IgG ELISA kits (Invitrogen), following the manufacturer's instructions.

## Murine pneumonia challenge

All animal procedures were conducted according to the guidelines of the Emory University Institutional Animal Care and Use Committee (IACUC), under approved protocol number PROTO 201700441. The study was carried out in strict accordance with established guidelines and policies at Emory University

School of Medicine, and recommendations in the Guide for Care and Use of Laboratory Animals, as well as local, state, and federal laws. *P. aeruginosa* strain PA103 (ATCC 29260, sputum isolate) was grown on Difco Pseudomonas isolation agar for 16–18 hr at 37 °C and suspended in PBS. Inocula were adjusted spectrophotometrically to obtain the desired challenge dose in a volume of 10–20 µl. Eight-to 10-wk-old female BALB/c mice (Jackson Laboratories, Bar Harbor, ME) were anesthetized by intraperitoneal injection of 0.2 ml of a cocktail of ketamine (100 mg/ml) and xylazine (5 mg/ml) and intranasally instilled with approximately $10^5$ CFU *P. aeruginosa* PA103 (in 10–20 µL of PBS) as described (*Moustafa et al., 2023*). At 15 min post-infection, monoclonal antibodies or PBS were delivered via the same route in a 20 µl volume. Mice were euthanized at 24 hr post-infection, and whole lungs were collected aseptically, weighed, and homogenized for 20 s in 1 ml of PBS, followed by serial dilution onto Pseudomonas isolation agar, and plated for CFU enumeration.

## Materials availability

A patent application based on this research, entitled 'Protective monoclonal antibodies to *Pseudomonas aeruginosa*,' has been filed by the University of Washington. Materials will be made available to the scientific community in accordance with the policies and procedures of the University of Washington.

## Acknowledgements

Jennifer Haddock provided indispensable administrative support. We thank the people with cystic fibrosis for their participation in this study.

## Additional information

### Competing interests

Malika Hale, Kennidy K Takehara, Christopher D Thouvenel, David J Rawlings, Marion Pepper: M Hale, KK Takehara, CD Thouvenel, M Pepper and DJ Rawlings are listed as inventors on a patent based on this work, entitled "Protective monoclonal antibodies to Pseudomonas aeruginosa," which has been filed by the University of Washington as PCT/US2024/057782. The other authors declare that no competing interests exist.

### Funding

| Funder | Grant reference number | Author |
| --- | --- | --- |
| National Institutes of Health | P30DK089507 | Sharon McNamara Ronald L Gibson Marion Pepper |
| Cystic Fibrosis Foundation | SINGH19R0 | Sharon McNamara Ronald L Gibson |
| National Institutes of Health | 5T32GM007266 | Malika Hale |
| Seattle Children's Foundation | Tom Hansen Investigator for Pediatric Innovation Endowment | David J Rawlings |

The funders had no role in study design, data collection and interpretation, or the decision to submit the work for publication.

### Author contributions

Malika Hale, Conceptualization, Formal analysis, Validation, Investigation, Visualization, Writing – original draft, Writing – review and editing; Kennidy K Takehara, Dina A Moustafa, Jason Netland, Conceptualization, Formal analysis, Validation, Investigation, Visualization, Writing – review and editing; Christopher D Thouvenel, Conceptualization, Data curation, Formal analysis, Validation, Investigation, Visualization, Writing – review and editing; Andrea Repele, Formal analysis, Validation, Investigation, Writing – review and editing; Mary F Fontana, Conceptualization, Formal analysis,

Validation, Investigation, Writing – review and editing; Sharon McNamara, Resources, Supervision, Writing – review and editing; Ronald L Gibson, Resources, Supervision, Funding acquisition, Project administration, Writing – review and editing; Joanna B Goldberg, Conceptualization, Supervision, Funding acquisition, Project administration, Writing – review and editing; David J Rawlings, Conceptualization, Formal analysis, Supervision, Funding acquisition, Writing – original draft, Project administration, Writing – review and editing; Marion Pepper, Conceptualization, Formal analysis, Supervision, Funding acquisition, Project administration, Writing – review and editing

### Author ORCIDs
Malika Hale ⓘ https://orcid.org/0000-0002-0822-7141
Mary F Fontana ⓘ https://orcid.org/0000-0003-3630-181X
David J Rawlings ⓘ https://orcid.org/0000-0002-0810-1776
Marion Pepper ⓘ https://orcid.org/0000-0001-7278-0147

### Ethics
Human subjects were recruited and informed consent was obtained following strict protocols approved by the Seattle Children's Institutional Review Board, under protocol number SCH#10325. All animal procedures were conducted according to the guidelines of the Emory University Institutional Animal Care and Use Committee (IACUC), under approved protocol number PROTO 201700441. The study was carried out in strict accordance with established guidelines and policies at Emory University School of Medicine, and recommendations in the Guide for Care and Use of Laboratory Animals, as well as local, state, and federal laws.

Joint Public Review: https://doi.org/10.7554/eLife.98851.3.sa1
Author response https://doi.org/10.7554/eLife.98851.3.sa2

## Additional files

### Supplementary files
MDAR checklist

### Data availability
All sequencing data have been submitted to the NCBI for research use under BioProject identifier PRJNA1231685.

The following dataset was generated:

| Author(s) | Year | Dataset title | Dataset URL | Database and Identifier |
|---|---|---|---|---|
| Hale M, Takehara KK, Thouvenel CD, Moustafa DA, Repele A, Fontana MF, Gibson RL, Goldberg JB, Rawlings DJ, Pepper M | 2025 | Monoclonal antibodies derived from B cells in subjects with cystic fibrosis | https://www.ncbi.nlm.nih.gov/bioproject/PRJNA1231685 | NCBI BioProject, PRJNA1231685 |

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
