## [Editor Report · eLife Assessment]

Treatment of *Pseudomonas aeruginosa* (PA) infections is challenging because of intrinsic and acquired antibiotic resistance to most antibiotic drug classes. Therefore, by using donor B cells in subjects with cystic fibrosis who undergo intermittent or chronic airway PA infections, the authors aimed to isolate B-cell receptors against PA virulence factors and examined their biological activities. The data are **solid** and the protective antibodies identified in this study could be **useful** for protection against PA.

---

## [Referee Report · Joint Public Review]

Summary:

This study presents a strategy to efficiently isolate PcrV-specific BCRs from human donors with cystic fibrosis who have/had *Pseudomonas aeruginosa* (PA) infection. Isolation of mAbs that provide protection against PA may be a key to developing a new strategy to treat PA infection as the PA has intrinsic and acquired resistance to most antibiotic drug classes. Hale et al. developed fluorescently labeled antigen-hook and isolated mAbs with anti-PA activity. Overall, the authors' conclusion is supported by solid data analysis presented in the paper. Four of five recombinantly expressed PcrV-specific mAbs exhibited anti-PA activity in a murine pneumonia challenge model as potent as the V2L2MD mAb (equivalent to gremubamab). However, therapeutic potency for these isolated mAbs is uncertain as the gremubamab has failed in Phase 2 trials. Clarification of this point would greatly benefit this paper.

Strengths:

(1) High efficiency of isolating antigen-specific BCRs using an antigenic hook.

(2) The authors' conclusion is supported by data.

Weaknesses:

Although the authors state that the goal of this study was to generate novel protective mAbs for therapeutic use (P12; Para. 2), it is unclear whether PcrV-specific mAbs isolated in this study have therapeutic potential better than the gremubamab, which has failed in Phase 2 trials. Four of five PcrV-specific mAbs isolated in this study reduced bacterial burdens in mice as potent as, but not superior to, gremubamab-equivalent mAb. Clarification of this concern by revising the text or providing experimental results that show better potential than gremubamab would greatly benefit this paper.

---

## [Author Response]

The following is the authors’ response to the original reviews.

**Joint Public Review:**
Summary:This study presents a strategy to efficiently isolate PcrV-specific BCRs from human donors with cystic fibrosis who have/had *Pseudomonas aeruginosa* (PA) infection. Isolation of mAbs that provide protection against PA may be a key to developing a new strategy to treat PA infection as the PA has intrinsic and acquired resistance to most antibiotic drug classes. Hale et al. developed fluorescently labeled antigen-hook and isolated mAbs with anti-PA activity. Overall, the authors' conclusion is supported by solid data analysis presented in the paper. Four of five recombinantly expressed PcrV-specific mAbs exhibited anti-PA activity in a murine pneumonia challenge model as potent as the V2L2MD mAb (equivalent to gremubamab). However, therapeutic potency for these isolated mAbs is uncertain as the gremubamab has failed in Phase 2 trials. Clarification of this point would greatly benefit this paper.Strengths:(1) High efficiency of isolating antigen-specific BCRs using an antigenic hook.(2) The authors' conclusion is supported by data.Weaknesses:Although the authors state that the goal of this study was to generate novel protective mAbs for therapeutic use (P12; Para. 2), it is unclear whether PcrV-specific mAbs isolated in this study have therapeutic potential better than the gremubamab, which has failed in Phase 2 trials. Four of five PcrV-specific mAbs isolated in this study reduced bacterial burdens in mice as potent as, but not superior to, gremubamab-equivalent mAb. Clarification of this concern by revising the text or providing experimental results that show better potential than gremubamab would greatly benefit this paper.

The authors thank the reviewer for their thoughtful positive assessment. As noted by the reviewer, the studies described here, which were performed in mice, show that our MBC-derived mAbs are as effective as V2L2MD, a mAb that is one component of the gremubamab bi-specific. However, key theoretical strengths of MBC-derived mAbs (reduced immunogenicity, full participation in effector functions) are not easily tested in mice. We have clarified and expanded our discussion of these points in our revised manuscript, particularly in the Discussion paragraph 4.

**Recommendations for the authors:**

**Reviewer #1 (Recommendations for the authors):**
Page 8. Using improved methods that enhanced the efficiency and depth of sequencing (manuscript in preparation...). This method is not provided in detail. The authors should provide a detailed method (as a preprint on a public database or described in the method section).

We thank the reviewers for their interest in the details of the specific methods for single cell B cell receptor sequencing. We regret that the manuscript is still in preparation. In fact, our current methods section provides much more detail about sequencing methods than is customarily supplied by authors mAb development papers. However, we understand the frustration and will remove our citation of our manuscript in preparation in our revised manuscript.